# Unveiling Players’ Perceptions of Mother- and Father-Initiated Motivational Climates and Fear of Failure in Youth Male Team Sports

**DOI:** 10.3390/sports12090244

**Published:** 2024-09-04

**Authors:** Patrícia Coutinho, Cristiana Bessa, Cláudia Dias, Isabel Mesquita, António M. Fonseca

**Affiliations:** CIFI2D, Centre of Research, Education, Innovation and Intervention in Sport, Faculty of Sport, University of Porto, Rua Dr. Plácido Costa, 91, 4200-450 Porto, Portugal; cbessa@fade.up.pt (C.B.); cdias@fade.up.pt (C.D.); imesquita@fade.up.pt (I.M.); afonseca@fade.up.pt (A.M.F.)

**Keywords:** parent involvement, fear of failure, team sports, youth sport

## Abstract

The aim of this study was to examine the relationship between perceived mother- and father-initiated motivational climates and players’ fear of failure in youth male team sports. A sample of 336 youth male players from five team sports (basketball, football, handball, volleyball, and water polo) completed the Parent-Initiated Motivational Climate Questionnaire-2 and the Performance Failure Appraisal Inventory. The results showed that perceived mother- and father-initiated motivational climates were related to fear of failure predispositions. While a mastery orientation (perceived learning-enjoyment climate) had a low association with fear of failure, an ego orientation (perceived worry-conducive and success-without-effort climates) was highly related to fear of failure. Father-initiated climates had stronger associations with fear of failure than mother-initiated ones, revealing that mothers and fathers may have different influences when considering the developmental origins of fear of failure. The relationships between mother- and father-initiated motivational climates and fear of failure varied according to the type of sport, with basketball, football, and volleyball presenting stronger associations. The dimensions “Fear of important others losing interest” and “Fear of upsetting important others” presented the highest explained variance in all sports when predicted by the father-initiated motivational climate. The findings can inform important evidence-based guidelines and recommendations for parents, coaches, and organizations, enabling them to create supportive environments that aid athletes in developing the necessary psychological skills for long-term success and well-being.

## 1. Introduction

In the realm of youth sports, parental influence has been widely recognized as a significant factor shaping athletes’ psychological experiences and developmental outcomes [1,2,3,4]. Specifically, parent-initiated motivational climate, defined as the socioemotional environment created by parents that influences athletes’ motivation, perceptions, and behaviors, has garnered substantial attention in the last two decades among sport psychology researchers [5,6,7,8,9].

Nicholls’ Achievement Goal Theory (AGT) [10] is one of the most widely used theoretical frameworks for underpinning the study of parental influences throughout athletes’ sport participation. It posits that individuals’ motivation and behavior in achievement settings like sport are influenced by the type of goals they pursue. Accordingly, parents can create mastery-oriented or performance-oriented climates [10,11]. A mastery-oriented climate focuses on personal improvement, skill development, and effort, fostering a mastery goal orientation, while a performance-oriented climate emphasizes competition, winning, and social comparisons, promoting a performance goal orientation [12,13].

Studies have consistently demonstrated the positive impact of a parent-initiated mastery-oriented motivational climate on an athlete’s overall development [2,14]. Athletes raised in such a climate tend to exhibit higher levels of intrinsic motivation, task enjoyment, persistence, and adaptive achievement behaviors [15,16,17,18]. They are more likely to seek challenges, persist in the face of setbacks, and focus on personal growth and skill improvement [19,20,21]. On the other hand, an ego-oriented motivational climate created by parents is often associated with high levels of extrinsic motivation driven by external rewards [18,22], higher levels of anxiety [18], and perfectionistic cognitions [23]. The excessive focus on performance outcomes can also hinder athlete’s skill development as they prioritize short-term success over long-term growth and learning [19,24].

The motivational climate created by parents has been also associated with the way athletes perceive specific sporting situations as threatening, challenging, or stressful (e.g., defeat, performance slumps, performing in a crucial competition) [14,25]. As such, the motivational climate generated by parents may have an important impact on athletes’ fear of failure predispositions [25]. Fear of failure has been defined as a multifaceted psychological construct characterized by individuals’ cognitive, emotional, and behavioral responses to the perceived threat of failing to meet personal performance expectations or desired outcomes in sports-related endeavors [26,27,28,29]. It encompasses the apprehension, worry, and distress associated with the possibility of experiencing unsuccessful performances, negative evaluations, or falling short of self-imposed standards or societal norms in the sporting domain [26,29,30,31].

Conroy and colleagues [27] proposed a model suggesting that fear of failure comprises five beliefs concerning the repercussions of failure which are linked to threat appraisal and feared outcomes. These beliefs include the following: (i) experiencing shame and embarrassment upon failure, related to beliefs about self-presentational failure and personal diminishment; (ii) having an uncertain future, linked to beliefs of losing future opportunities; (iii) devaluing one’s self-esteem, associated with beliefs about having reduced control over one’s performance; (iv) important others losing interest, related to beliefs about losing social value; and (v) upsetting important others, related to beliefs about others’ disapproval and loss of affection following failure. Therefore, Conroy and colleagues [27] conceptualized fear of failure as a construct shaped by one’s interpersonal interactions and reflective of the desire to avoid a decline in one’s evaluation.

Fear of failure is therefore considered to be a motive disposition that has some characteristics of a stable trait and, as such, its manifestation could be activated by situational factors, such as parental influence [25,31,32]. Considering that sport exists in a social context in which athletes’ performances are evaluated by the self and by important others, parents and the motivational climates created by them play an important role in young athletes’ experiences in youth sport [22,33], particularly in the way they experience and interpret fear of failure in this specific domain [16,25].

Nonetheless, the influences of parent-initiated motivational climates on athletes’ fear of failure have received far less academic attention within the literature [16]. To the best of our knowledge, only the study of Sagar and Lavallee [25] has examined the relationship between these two constructs, highlighting a clear positive association between a parentally driven ego-oriented motivational climate and fear of failure [25]. The authors found that parents’ punitive and controlling behaviors as well as their high expectations for achievement contributed to athletes’ fear of failure [25]. In contrast, parental mastery-oriented and autonomy-supportive climates have been negatively associated with fear of failure [25]. Such evidence supports the notion that parents play a crucial role in athletes’ experiences in youth sport [4,16,18] and may have an even more significant role than other social agents (e.g., coaches) in the way athletes interpret and deal with fear of failure predispositions.

Despite such evidence, the intricate interplay between the parent-initiated motivational climate and athletes’ fear of failure remains largely understudied. The few existing studies on this topic have focused on youth athletes in general [31], failing to provide specific knowledge about the concrete ages for this developmental period, such as adolescence. This developmental stage (adolescence) is a critical period for shaping the formation of identity, values, and beliefs, including those related to sports participation [34], so the understanding about the influence of parental motivational climate on fear of failure during this specific phase is pivotal. Furthermore, the existing studies have mainly explored samples of youth athletes from individual sports [25], which focuses knowledge on these specific populations. 

Considering that fear of failure experiences may differ across various sports and cultural contexts [25], it is important to explore other specific sporting contexts like team sports. These are considered late specialization sports [35,36], comprising specific developmental features during adolescence and consequently leading to a range of psychological and emotional experiences and outcomes. This is particularly pertinent when considering different types of team sports, due to their varying characteristics and approaches. Here, parents play a significant role in both supporting and hindering those experiences, which consequently could have a detrimental effect on athletes’ fear of failure predispositions. Additionally, researchers have identified differences in male and female athletes’ perceptions of fear of failure [16,25], making it valuable to examine single-gender samples of athletes for a more specific understanding of this issue. Finally, although studies on this topic have examined the motivational climates initiated by both fathers and mothers separately, the truth is that this evidence is interpreted all together, failing to provide a detailed understanding about the influence of each of these agents on athletes’ fear of failure predispositions. Examining the motivational climate provided by mothers versus fathers separately is therefore crucial for advancing knowledge in this research field. Such knowledge is important for providing important evidence-based guidelines and recommendations for parents, coaches, and organizations to create more supportive and growth-oriented sporting environments.

Therefore, the purpose of this study was to examine the relationships between perceived mother- and father-initiated motivational climates and players’ fear of failure in youth male team sports. Specifically, we sought to understand the following: (1) to what extent a mastery or ego orientation may impact fear of failure dispositions differently in these players; and (2) if these influences may vary according to the type of team sport considered (basketball, handball, football, volleyball, and water polo).

## 2. Materials and Methods

### 2.1. Participants and Procedures

The data for this paper are derived from the “In Search of Excellence in Sport—a mixed-longitudinal study in young male athletes” (INEX) research project. A detailed description of the INEX study design can be found elsewhere [37]. The present study examined 336 youth male players from five team sports (94 basketball players, 119 football players, 35 handball players, 58 volleyball players and 30 water polo players), with an age range of between 13 and 16 years (overall players’ average age of 14.14 ± 0.82 years). Purposive and convenience sampling criteria were used to select the participants for this study [38]. They were chosen for the study because they were considered “information-rich” [39] since they (i) had at least one year of specialized training and competitive experience in one of the team sports considered in this study, (ii) fell within the age range intended for this study (i.e., adolescence), which is a critical developmental period characterized by significant physical, cognitive, social, and emotional changes that shape the formation of identity, values, and beliefs, including those related to sports participation [34], and (iii) had the ability and willingness to participate in the study. The 336 players were selected from clubs within the respective local sporting associations. The coaches and club coordinators chose the participants based on the aforementioned criteria. The unequal distribution of the participants among the sports reflects the popularity of the sports under analysis and the variation in participants according to this factor.

Ethical approval was granted by the Research Ethics Committee of the first author’s institution, guided by the Declaration of Helsinki (process number CEFADE 13.2017). Participants and their parents were informed about the study’s purpose, their scope of involvement, and of their right to withdraw at any time. Data confidentiality was ensured and informed consent forms were signed by both parents and participants.

For data collection procedures, participants engaged in the completion of a questionnaire within a controlled environment (a quiet classroom setting) under the direct supervision of a designated research team member. All 336 players responded to the questionnaire. Although participants concurrently completed the questionnaire in a group setting, everyone worked independently, with approximately 30 participants per group. Prior to commencement, participants were reassured of the confidentiality and anonymity of their responses, accompanied by comprehensive verbal instructions. Clarifications regarding the questions and variables under analysis were offered as required. The entire questionnaire completion process spanned approximately 30 to 40 min.

### 2.2. Measurement Instruments

The Parent-Initiated Motivational Climate Questionnaire-2 (PIMCQ-2p). The Portuguese version of the PIMCQ-2p [40] was used to assess the athletes’ perceptions of the motivational climate created by both fathers and mothers. The questionnaire consists of three subscales, one reflecting a mastery orientation (learning-enjoyment climate: 8 items, e.g., “I feel that my mother/father is most satisfied when I learn something new in sport”) and two reflecting ego orientation (worry-conducive climate: 5 items, e.g., “I feel that my mother/father makes me worried about losing in sport”, and success-without-effort climate: 5 items, e.g., “I feel that may mother/father is most satisfied when I succeed without effort in sport”). Responses are recorded on a 5-point Likert scale ranging from “strongly disagree” (1) to “strongly agree” (5). Santana and colleagues [40] reported the psychometric properties of the PIMCQ-2p, demonstrating the feasibility of the instrument. In the present sample, the internal consistency was acceptable (Cronbach’s alpha = 0.875 to 0.885 for the mastery climate scale and Cronbach’s alpha = 0.756 to 0.815 for the ego climate scale).

The Performance Failure Appraisal Inventory (PFAI). The Portuguese validated version of the PFAI was used [41] to assess fear of failure in the sporting context. The inventory includes 19 items grouped into five dimensions: Fear of Experiencing Shame and Embarrassment (5 items) (e.g., “When I am failing, it is embarrassing if others are there to see it”), Fear of Devaluing One’s Self-estimate (3 items) (e.g., “When I am failing, it is often because I am not smart enough to perform successfully”), Fear of Having an Uncertain Future (3 items) (e.g., “When I am failing, I believe that my plans for the future will change”), Fear of Important Others Losing Interest (4 items) (e.g., “When I am not succeeding, some people are not interested in me anymore”), and Fear of Upsetting Important Others (4 items) (e.g., “When I am failing, important others are disappointed”). The answers were collected on a Likert-type scale ranging from “do not believe at all” (1) to “believe 100% of the time” (5). Coutinho and colleagues [41] reported the psychometric properties of the PFAI, demonstrating the feasibility of the instrument. In the present sample, internal consistency was acceptable (Cronbach’s alpha = 0.601 to 0.883).

Data quality control was ensured through the following steps: (1) all questionnaires were administered by trained personnel from the authors’ faculty, and (2) data cleaning was conducted to correct possible data entry errors, identify potential outliers, and verify the basic assumptions of the statistical tests.

### 2.3. Data Analysis

Descriptive statistics were used to calculate frequencies, percentages, means, and standard deviation values. The requirements of normality and homogeneity of variance were examined through the Kolmogorov–Smirnov test and Levene’s test. Pearson’s product moment correlation coefficient analyses were firstly conducted to examine the relationship between parent-initiated motivational climates (by both mothers and fathers) and fear of failure. Cohen’s conventions were used to determine correlation coefficient size, where r = 0.30 is considered small, r = 0.50 is considered moderate, and r = 0.70 is considered large. Next, a multiple linear regression analysis was used to investigate the associations between parent-initiated motivational climates and fear of failure (in general and according to the type of sport). The assumptions of multiple linear regression, namely the linearity of the relationship between the independent variables and the dependent variable (graphical analysis), independence of residuals (Durbin–Watson test), normality of residuals (Kolmogorov–Smirnov test), multicollinearity (VIF and tolerance) and homoscedasticity (graphical analysis) were analyzed and found to be generally satisfied. Finally, a multivariate analysis of variance (MANOVA) was conducted to examine the differences among groups (i.e., basketball, handball, football, volleyball, and water polo players) on parent-initiated motivational climate and fear of failure. The multivariate effect of the independent variable was assessed using Pillai’s trace statistic, which provides an overall measure of the differences among the groups. Effect size values were determined by partial eta squared (η2P), considered as small (η2P < 0.06), moderate (0.06 ≤ η2P < 0.15), or large (η2P ≥ 0.15) [42]. Post hoc analyses were conducted using Bonferroni tests (Bonferroni adjusted alpha of *p* = 0.001). Statistical significance was set at *p* < 0.05

## 3. Results

Means, standard deviations, alpha coefficients, and Pearson product moment correlations for perceived mother- and father-initiated motivational climate and fear of failure are reported in Table 1. Significant relationships between the perceived mother-initiated motivational climate and fear of failure were observed, namely (1) a significant small and negative relationship between learning-enjoyment climate and the five dimensions of fear of failure (r = −0.115 to −0.271, all *p* < 0.05), (2) a significant small and positive relationship between the worry-conducive climate and the fear of important others losing interest (r = 0.108, *p* = 0.04), and (3) a significant moderate and positive relationship between the success-without-effort climate and the five dimensions of fear of failure (r = −0.341 to −0.433, all *p* < 0.001). Concerning the perceived father-initiated motivational climate, results showed (1) a significant small and negative relationship between the learning-enjoyment climate and the five dimensions of fear of failure (r = −0.145 to −0.312, all *p* < 0.05), (2) a significant moderate and positive relationship between the worry-conducive climate and the five dimensions of fear of failure (r = −0.425 to −0.593, all *p* < 0.001), and (3) a significant small and positive relationship between the success-without-effort climate and the five dimensions of fear of failure (r = −0.347 to −0.465, all *p* < 0.001).

Multiple linear regression analysis was used to assess associations between perceived mother- and father-initiated motivational climate and fear of failure. The model is shown in Table 2. The total variance explained by the model when considering the influence of the motivational climates created by fathers and mothers on the five dimensions of fear of failure ranged between 14% and 43%. Specifically, the perceived climate created by fathers had a greater contribution to fear of failure (22% to 43%) when compared to the perceived climate generated by mothers (14% to 26%). The dimension “Fear of important others losing interest” was the dimension with a higher percentage of explained variance in the motivational climate generated by the father (43%) and mother (26%). The learning-enjoyment climate (both mother and father) was a negative predictor of the five dimensions of fear of failure.

Descriptive and inferential statistics for perceived mother- and father-initiated motivational climate and fear of failure according to the type of sport are presented in Table 3. Significant differences were found between participants with players differing in the perceptions of mother-initiated learning-enjoyment climate (F_(4337)_ = 2.808, *p* = 0.026, η^2^_P_ = 0.033), mother-initiated worry-conducive climate (F_(4337)_ = 3.016, *p* = 0.018, η^2^_P_ =0.035), mother-initiated success-without-effort climate (F_(4337)_ = 3.674, *p* = 0.006, η^2^_P_ = 0.042) and father success-without-effort climate (F_(4337)_ = 3.623, *p* = 0.007, η^2^_P_ = 0.042). Specifically, volleyball players perceived a more learning-enjoyment-based climate provided by mothers when compared to water polo players (*p* = 0.009), but also a more worry-conducive climate provided by mothers when compared to basketball players (*p* = 0.022). Basketball players perceived a less success-without-effort climate provided by mothers and fathers when compared to handball players (*p* = 0.031 and *p* = 0.038, respectively) and volleyball players (*p* = 0.040 and *p* = 0.013, respectively). Players did not differ significantly concerning fear of failure predispositions.

Multiple linear regression analysis was used to assess associations between perceived mother- and father-initiated motivational climate and fear of failure according to the type of sport. The model is shown in Table 4 and suggests that perceived mother- and father-initiated motivational climate is a significant predictor of the five dimensions of fear of failure in all sports. The exception was observed for the mother-initiated motivational climate in handball and water polo players, as shown in Table 4. The overall model accounted for 14–33% of variance in basketball, 15–60% of variance in handball, 10–33% of variance in football, 22–59% of variance in volleyball and 16–68% of variance in water polo. A mastery orientation climate (learning-enjoyment climate) initiated by both mothers and fathers was a negative predictor of the five dimensions of fear of failure in all sports (β values ranging from −0.015 to −0.479). In contrast, an ego orientation climate (worry-conducive and success-with effort climates) was a positive predictor in the majority of sports (β values ranging from 0.005 to 0.760). The dimensions “Fear of important others losing interest” and “Fear of upsetting important others” presented the highest variance in all sports when predicted by the father-initiated motivational climate (33–68% and 24–48%, respectively).

## 4. Discussion

The main purpose of the present study was to examine relationships between perceived mother- and father-initiated motivational climates on fear of failure in youth male team sport players. Globally, this study demonstrated that the perceived motivational climate generated by parents (created by both fathers and mothers) have significant associations with fear of failure in the players examined in this study. Moreover, the type of motivational climate orientation (i.e., mastery or ego orientation) impacted fear of failure predispositions differently. While father- and mother-initiated mastery orientations (i.e., perceived learning-enjoyment climates) were less associated with fear of failure, an ego orientation (i.e., perceived worry-conducive and success-without-effort climates) generated by both fathers and mothers were highly related to fear of failure. These global results are aligned with previous research indicating that when parents promote a mastery-oriented motivational climate, athletes are less likely to demonstrate fear of failure in sport [25]. Considering that the roots of fear of failure are laid down early in childhood development [25], a mastery-oriented climate nurtures self-regulation skills and encourages children to adopt a learning-oriented approach, leading therefore to a reduced predisposition to perceiving threat and feeling anticipatory shame in evaluative situations [14]. On the other hand, an ego-oriented climate over time may lead players to experience worry and negative emotions (e.g., anxiety, shame, and sadness) and consequently learn to associate failure with this type of parental environment [14,18,27]. Such findings are of the utmost importance, since sport practitioners (e.g., coaches and sport psychologists) should provide guidance for both parents and players regarding these issues.

Remarkably, the results of this study are novel in so far as they show that father-initiated motivational climates have stronger associations with player’s fear of failure predisposition when compared to mother-initiated motivational climates. Specifically, a father-initiated worry-conducive climate and a father-initiated success-without-effort climate were more strongly related to the five dimensions of fear of failure than the equivalent mother-initiated climate dimensions. To the best of our knowledge, no studies have yet reported such differences between father- and mother-initiated motivational climate and its influence on fear of failure within the sport context. A possible explanation regarding why players’ predispositions to fear of failure show stronger relations to the dimensions of father-initiated climate might be that fathers have a particular significant role as caregivers through the adolescent years, particularly for male athletes. Côté [34] suggested that emotional parental support remains very high (and may even increase) during the adolescent years, suggesting that parents still have an important role during this phase of development. Given such involvement and support, it is likely that fathers continue to influence player’s sport participation and particularly the way they perceive the threat of failing to meet personal performance expectations or desired outcomes in sports. Here, specific orientations provided by sport psychologists regarding this issue may help parents to align their behaviors and attitudes according to what is expected within the sporting environment.

Furthermore, the results of this study are also innovative to the extent that parental ego orientation climates (i.e., perceived worry-conducive and success-without-effort climates) initiated by both mothers and fathers are strongly associated with one particular dimension of fear of failure, namely the “fear of important others losing interest”. Considering that the adolescent years are an important phase of socialization, players can fear others’ negative judgement, avoiding mistakes and failure, and consequently becoming fearful of failure [25]. Over time, players may have learnt to associate failure with feeling guilty for disappointing their significant others (e.g., coaches, peers, and parents). This, in turn, can be detrimental to the player’s performance, well-being, motivation and self-esteem, compromising therefore their sport participation in the long-term [14,16].

Of special interest, this study reveals that the perceived mother- and father-initiated motivational climate was strongly related to the five dimensions of fear of failure in most of the team sports examined. The exception was observed for the mother-initiated motivational climate in handball and water polo players, where no significant associations were found. This could be related to the fact that football, basketball, and volleyball are very popular and expressive sports in the country in which this study was developed, presenting a demanding competitive environment already in the specialization phase (i.e., adolescent years), which could imply different parental involvement, behavior, and influences. However, fear of failure is a multifaceted psychological construct influenced by several situational factors [20,21,25,32], so the motivational climate created by parents, although important, is not the only factor that may determine fear of failure dispositions in sport. This means that other important influences may have determined fear of failure in those players. Research has shown that the motivational climates generated by coaches and teammates also have an important impact in the way athletes perceive fear of failure [43]. Such evidence could explain the fact that the dimensions “Fear of important others losing interest” and “Fear of upsetting important others” presented the highest explained variance in all sports when predicted by the father-initiated motivational climate. Accordingly, the adolescent years are an important phase of socialization, which may lead players to fear negative judgements from significant others (e.g., parents, coaches, and peers/teammates), becoming therefore fearful of failure in their sporting contexts [25]. This finding is of the utmost importance, since it could help sport practitioners (e.g., sport psychologists) understand that athletes are not all equal and provide concrete and assertive guidance to them according to their sporting context. Further studies are, however, needed to explore such issues in different sporting contexts and cultures and provide a deeper perspective on the idiosyncrasies of parent-initiated motivational climates and their associations with fear of failure [2,20,21].

### Limitations, Directions for Future Research, and Practical Implications

Despite the aforementioned novel findings, a number of limitations in this study should be addressed. The present study used quantitative methodology, based on reliable and valid instruments widely reported in the literature. However, such methodology only reflects interpretations of participants’ reports and perceptions. While these perceptions are an important source of information, they do need to be triangulated with other objective data to provide a rich understanding of the influences of the motivational climate initiated by parents on fear of failure predispositions. This is particularly important because self-report measures, like those used in this study, are susceptible to social desirability effects (i.e., participants may provide answers that are more socially acceptable than their true opinions or behaviors). In this context, employing qualitative research methods such as in-depth interviews, focus groups, participant observation, reflective notetaking, action research, and ethnographic studies can provide a more profound perspective. This approach aims to enhance the understanding of the study’s topic and its potential relationships with other variables.

Additionally, this study has a cross-sectional design examining perceived parent-initiated motivational climates in a certain moment. Considering that the motivational climate may vary over time [9,44], future research should benefit from a prospective longitudinal and/or experimental design. Furthermore, this study examined a sample of adolescent players, which provided a concrete understanding of a specific phase of development. Further investigations should study athletes from different age groups to understand the association between the motivational climates generated by parents and fear of failure considering different phases of development (e.g., childhood and adulthood). Similarly, further studies should also explore the perspectives of other participant groups, namely athletes from different sporting contexts, cultures, and social realities, as well as female athletes, to provide a broader and deeper perspective on this topic [2,20,21].

Finally, this study presented solely the players’ perceptions regarding perceived parent-initiated motivational climates and their own fear of failure predispositions. Researchers should also examine the perceptions of parents about the motivational climates created by them and how these influence athletes’ fear of failure within the sporting context. Such research will be beneficial as it can further explain the link between parental fear of failure and its impact on the way they behave (within the motivational climates generated by them), and the developmental origins of fear of failure in their children [25].

Ultimately, elucidating the connection between parental influence and an athlete’s fear of failure holds practical implications for coaches, sport psychologists, and strategies involving parental involvement in sports. Insights derived from these studies could aid in the design of more effective training programs, particularly through parental education initiatives delivering targeted interventions related to motivational orientations, communication, and support. These interventions may be designed to address specific long-term developmental outcomes, thereby enhancing the overall impact on athletes’ fear of failure and performance.

## 5. Conclusions

The findings of this study reinforce the idea that the motivational climate established by parents within the sporting environment has important psychological consequences for young athletes. Particularly, this study demonstrated that perceived mother- and father-initiated motivational climates had important contributions to fear of failure predispositions in adolescent male team sport players. However, key insights arise from this investigation, since father-initiated motivational climates had stronger associations with fear of failure, revealing that mothers and fathers may have different roles and influences when considering the developmental origins of fear of failure. Thus, it is of paramount importance that further studies consider the motivational climates generated by mothers and fathers separately to provide a deeper perspective on the idiosyncrasies of such constructs. A cornerstone of this is the need to consider specific developmental periods or age groups (i.e., childhood, adulthood, start of sporting participation, and specialization phase in sport), since this study revealed important insights for a particular age group and phase of sporting development. Finally, the innovative findings of this study showed that the type of sport could have an important influence when interpreting the developmental origins of fear of failure. In forthcoming studies, it will be imperative to consider different sporting contexts and cultures, as well as the context of female athletes, in pursuit of a more nuanced understanding of the relationships between parental motivational climate and fear of failure in sport. Research of this nature can yield important guidelines and recommendations for practice, especially for parents, coaches, and organizations, enabling them to create supportive environments that aid athletes in developing the necessary psychological skills for long-term success and well-being. Also, the results of this study emphasize the significance of formulating and executing parental education programs. Such programs serve as valuable tools to aid parents in comprehending, interpreting, and guiding their behaviors within the sporting context.

## Figures and Tables

**Table 1 sports-12-00244-t001:** Descriptive statistics, alpha coefficients, and Pearson product moment correlations for parent-initiated motivational climate (mother and father) and fear of failure.

		Parent-Initiated Motivational Climate						M ± Sd	α
		1	2	3	4	5	6		
Fear of Failure	Experiencing Shame and Embarrassment	−0.148 **	0.066	0.341 ***	−0.176 **	0.425 ***	0.347 ***	2.56 ± 0.9	0.830
	Devaluing One’s Self-estimate	−0.191 ***	−0.001	0.377 ***	−0.232 ***	0.467 ***	0.365 ***	2.18 ± 1.0	0.720
	Having an Uncertain Future	−0.115 *	0.036	0.382 ***	−0.145 **	0.463 ***	0.364 ***	2.20 ± 1.0	0.601
	Important Others Losing Interest	−0.268 ***	0.108 *	0.433 ***	−0.312 ***	0.593 ***	0.465 ***	1.97 ± 1.0	0.883
	Upsetting Important Others	−0.172 **	0.086	0.355 ***	−0.201 ***	0.565 ***	0.372 ***	2.16 ± 0.9	0.751
	M ± Sd	4.19 ± 0.7	2.04 ± 0.9	2.16 ± 1.0	4.26 ± 0.7	2.04 ± 0.9	2.15 ± 1.0		
	α	0.885	0.815	0.756	0.875	0.795	0.786		

Parent-initiated motivational climate: 1 = Learning-enjoyment climate (mother), 2 = Worry-conducive climate (mother), 3 = Success-without-effort climate (mother), 4 = Learning-enjoyment climate (father), 5 = Worry-conducive climate (father), 6 = Success-without-effort climate (father). *** *p* < 0.001, ** *p* < 0.01 and * *p* < 0.05 for statistical significance.

**Table 2 sports-12-00244-t002:** Multiple linear regression analysis for parent-initiated motivational climate (mother and father) and fear of failure.

		Fear of Failure														
		Experiencing Shame and Embarrassment			Devaluing One’s Self-Estimate			Having an Uncertain Future			Important Others Losing Interest			Upsetting Important Others		
		B	SE	β	B	SE	β	B	SE	β	B	SE	β	B	SE	β
PIMC mother	LEC	−0.194	0.073	−0.135	−0.260	0.073	−0.184	−0.156	0.076	−0.103	−0.358	0.068	−0.250	−0.216	0.069	−0.157
	WCC	0.047	0.053	0.045	−0.029	0.053	−0.027	0.019	0.055	0.017	0.076	0.049	0.074	0.063	0.050	0.063
	SWEC	0.343	0.052	0.336	0.387	0.052	0.373	0.405	0.054	0.379	0.431	0.048	0.425	0.340	0.049	0.349
R^2^		0.137			0.176			0.158			0.259			0.156		
F		17.748			23.765			20.830			38.958			20.619		
*p*		<0.001 *			<0.001 *			<0.001 *			<0.001 *			<0.001 *		
PIMC father	LEC	−0.152	0.073	−0.103	−0.229	0.72	−0.153	−0.097	0.075	−0.063	−0.310	0.061	−0.212	−0.142	0.064	−0.101
	WCC	0.326	0.059	0.316	0.368	0.058	0.350	0.392	0.060	0.362	0.451	0.049	0.439	0.478	0.051	0.485
	SWEC	0.175	0.053	0.182	0.174	0.053	0.179	0.181	0.055	0.180	0.220	0.045	0.231	0.113	0.047	0.124
R^2^		0.216			0.265			0.242			0.434			0.340		
F		30.609			40.060			35.583			85.195			57.395		
*p*		<0.001 *			<0.001 *			<0.001 *			<0.001 *			<0.001 *		

PIMC = parent-initiated motivational climate, LEC = Learning-enjoyment climate, WCC = Worry-conducive climate, SWEC = Success-without-effort climate; * *p* < 0.01 for statistical significance.

**Table 3 sports-12-00244-t003:** Descriptive and inferential statistics of parent-initiated motivational climate and fear of failure according to the type of sport.

Variables		Mean /Sd					F	*p*
		Handball	Basketball	Football	Water Polo	Volleyball		
Parent-Initiated Motivational Climate	Learning-enjoyment climate (mother)	4.21 ± 0.6	4.18 ± 0.6	4.21 ± 0.7	4.50 ± 0.5	4.01 ± 0.7	2.808	0.026 *
	Worry-conducive climate (mother)	1.95 ± 0.9	1.79 ± 0.7	2.10 ± 0.9	2.21 ± 1.1	2.26 ± 1.1	3.016	0.018 *
	Success-without-effort climate (mother)	2.45 ± 0.9	1.89 ± 0.8	2.15 ± 0.9	2.39 ± 1.1	2.34 ± 1.1	3.674	0.006 **
	Learning-enjoyment climate (father)	4.23 ± 0.6	4.21 ± 0.8	4.31 ± 0.6	4.45 ± 0.6	4.14 ± 0.7	1.505	0.200
	Worry-conducive climate (father)	1.93 ± 1.0	1.85 ± 0.8	2.15 ± 1.0	1.97 ± 1.0	2.21 ± 1.1	2.128	0.077
	Success-without-effort climate (father)	2.45 ± 1.0	1.87 ± 0.9	2.14 ± 0.9	2.27 ± 1.2	2.41 ± 1.2	3.623	0.007 **
Fear of Failure	Experiencing Shame and Embarrassment	2.26 ± 0.9	2.53 ± 0.9	2.53 ± 1.0	2.57 ± 1.0	2.85 ± 1.1	2.139	0.076
	Devaluing One’s Self-estimate	2.17 ± 1.0	2.04 ± 0.9	2.20 ± 1.0	2.26 ± 0.9	2.39 ± 1.1	1.191	0.315
	Having an Uncertain Future	2.03 ± 1.1	2.03 ± 0.9	2.33 ± 1.0	2.07 ± 1.1	2.40 ± 1.2	2.141	0.075
	Important Others Losing Interest	1.85 ± 1.0	1.93 ± 0.9	2.04 ± 1.0	1.62 ± 0.8	2.19 ± 1.1	2.053	0.087
	Upsetting Important Others	2.02 ± 0.9	2.09 ± 0.9	2.22 ± 0.9	1.92 ± 0.9	2.39 ± 1.1	1.818	0.125

** *p* < 0.01 and * *p* < 0.05 for statistical significance.

**Table 4 sports-12-00244-t004:** Multiple linear regression analysis for parent-initiated motivational climate (mother and father) and fear of failure according to the type of sport.

	Parent-Initiated Motivational Climate		Fear of Failure														
			Experiencing Shame and Embarrassment			Devaluing One’s Self-Estimate			Having an Uncertain Future			Important Others Losing Interest			Upsetting Important Others		
			B	SE	β	B	SE	β	B	SE	β	B	SE	β	B	SE	β
Basketball	Mother	LEC	−0.279	0.145	−0.188	−0.527	0.147	−0.342	−0.194	0.142	−0.131	−0.421	0.138	−0.283	−0.067	0.144	−0.045
		WCC	0.192	0.121	0.154	0.111	0.123	0.086	0.114	0.119	0.092	0.256	0.115	0.204	0.173	0.120	0.139
		SWEC	0.307	0.103	0.290	0.226	0.105	0.207	0.401	0.101	0.380	0.335	0.098	0.317	0.389	0.102	0.370
	R^2^		0.145			0.179			0.175			0.226			0.152		
	F		5.217			6.696			6.523			8.940			5.516		
	*p*		0.002 **			<0.001 **			<0.001 **			<0.001 **			0.002 **		
	Father	LEC	−0.187	0.117	−0.159	−0.427	0.117	−0.351	−0.119	0.107	−0.102	−0.249	0.103	−0.212	−0.041	0.107	−0.035
		WCC	0.258	0.119	0.219	0.114	0.120	0.094	0.292	0.110	0.249	0.390	0.106	0.331	0.453	0.110	0.387
		SWEC	0.180	0.102	0.175	0.184	0.102	0.174	0.386	0.094	0.376	0.305	0.090	0.296	0.255	0.094	0.250
	R^2^		0.142			0.195			0.274			0.329			0.241		
	F		5.064			7.421			11.570			15.037			11.077		
	*p*		0.003 **			<0.001 **			<0.001 **			<0.001 **			<0.001 **		
Handball	Mother	LEC	−0.460	0.255	−0.321	−0.435	0.286	−0.273	−0.655	0.294	−0.379	−0.667	0.250	−0.432	−0.662	0.233	−0.452
		WCC	−0.057	0.189	−0.054	0.015	0.213	0.013	0.141	0.219	0.112	0.076	0.186	0.067	0.195	0.173	0.182
		SWEC	0.255	0.172	0.261	0.297	0.194	0.273	0.320	0.199	0.271	0.353	0.169	0.335	0.280	0.158	0.281
	R^2^		0.170			0.148			0.236			0.231			0.333		
	F		1.908			1.624			2.883			4.110			4.662		
	*p*		0.151			0.206			0.053			0.016 *			0.009 **		
	Father	LEC	−0.049	0.270	−0.035	−0.189	0.234	−0.120	−0.465	0.321	−0.272	−0.480	0.200	−0.314	−0.310	0.237	−0.213
		WCC	0.323	0.213	0.343	0.687	0.184	0.657	0.305	0.252	0.269	0.517	0.157	0.510	0.450	0.187	0.468
		SWEC	0.168	0.179	0.191	0.029	0.155	0.029	0.124	0.212	0.117	0.174	0.132	0.184	0.115	0.157	0.127
	R^2^		0.156			0.488			0.181			0.601			0.375		
	F		2.906			10.831			3.278			16.535			7.198		
	*p*		0.052			<0.001 **			0.036 *			<0.001 **			<0.001 **		
Football	Mother	LEC	−0.105	0.125	−0.073	−0.163	0.130	−0.106	−0.238	0.125	−0.159	−0.403	0.113	−0.277	−0.152	0.116	−0.112
		WCC	0.113	0.090	0.110	−0.119	0.093	−0.108	−0.049	0.089	−0.046	0.068	0.081	0.065	−0.012	0.083	−0.013
		SWEC	0.329	0.093	0.308	0.442	0.096	0.388	0.458	0.093	0.411	0.468	0.084	0.434	0.382	0.086	0.378
	R^2^		0.104			0.170			0.183			0.304			0.141		
	F		5.697			8.050			10.040			17.184			7.637		
	*p*		0.001 *			<0.001 **			<0.001 **			<0.001 **			<0.001 **		
	Father	LEC	−0.178	0.146	−0.106	−0.050	0.152	−0.028	−0.108	0.146	−0.061	−0.416	0.132	−0.245	0.033	0.125	0.021
		WCC	0.321	0.097	0.321	0.439	0.101	0.412	0.475	0.097	0.455	0.338	0.088	0.335	0.532	0.083	0.563
		SWEC	0.119	0.099	0.115	0.107	0.103	0.097	0.062	0.099	0.057	0.213	0.090	0.204	0.032	0.085	0.033
	R^2^		0.167			0.207			0.238			0.331			0.314		
	F		9.075			11.526			13.583			20.918			19.423		
	*p*		<0.001 **			<0.001 **			<0.001 **			<0.001 **			<0.001 **		
Volleyball	Mother	LEC	−0.223	0.154	−0.172	−0.187	0.141	−0.153	0.140	0.161	0.104	−0.149	0.144	−0.116	−0.218	0.148	−0.179
		WCC	−0.128	0.119	−0.126	−0.091	0.109	−0.094	−0.013	0.125	−0.012	−0.044	0.111	−0.043	0.015	0.115	0.016
		SWEC	0.565	0.120	0.552	0.579	0.110	0.598	0.538	0.125	0.508	0.647	0.112	0.634	0.494	0.115	0.514
	R^2^		0.262			0.346			0.247			0.352			0.224		
	F		7.630			9.352			7.127			11.118			6.381		
	*p*		<0.001 **			<0.001 **			<0.001 **			<0.001 **			<0.001 **		
	Father	LEC	−0.024	0.175	−0.015	0.012	0.165	0.008	0.201	0.193	0.122	−0.138	0.137	−0.087	−0.381	0.146	−0.254
		WCC	0.368	0.138	0.360	0.360	0.130	0.371	0.311	0.152	0.293	0.633	0.108	0.621	0.411	0.115	0.427
		SWEC	0.294	0.123	0.325	0.273	0.116	0.318	0.274	0.135	0.292	0.211	0.096	0.234	0.280	0.103	0.329
	R^2^		0.337			0.343			0.253			0.594			0.478		
	F		10.472			10.764			7.318			28.340			18.099		
	*p*		<0.001 **			<0.001 **			<0.001 **			<0.001 **			<0.001 **		
Water Polo	Mother	LEC	−0.119	0.371	−0.061	−0.665	0.304	−0.378	−0.157	0.398	−0.076	−0.454	0.275	−0.277	−0.389	0.322	−0.227
		WCC	−0.045	0.165	−0.051	0.004	0.135	0.006	−0.112	0.177	−0.122	0.085	0.122	0.116	0.028	0.143	0.037
		SWEC	0.205	0.174	0.224	0.178	0.142	0.218	0.056	0.187	0.058	0.300	0.129	0.394	0.080	0.151	0.101
	R^2^		0.064			0.125			0.080			0.176			0.036		
	F		0.615			2.430			0.263			3.134			0.652		
	*p*		0.611			0.087			0.851			0.042 *			0.588		
	Father	LEC	−0.432	0.269	−0.269	−0.689	0.198	−0.479	−0.457	0.285	−0.270	−0.448	0.148	−0.334	−0.439	0.205	−0.314
		WCC	0.418	0.237	0.416	0.479	0.174	0.533	0.549	0.251	0.519	0.559	0.130	0.668	0.663	0.181	0.760
		SWEC	0.004	0.205	0.005	−0.033	0.151	−0.043	−0.145	0.217	−0.159	0.067	0.112	0.093	−0.238	0.156	−0.318
	R^2^		0.174			0.440			0.166			0.678			0.365		
	F		3.105			8.871			2.992			18.930			6.737		
	*p*		0.043 *			<0.001 **			0.048 *			<0.001 **			0.002 *		

LEC = Learning-enjoyment climate, WCC = Worry-conducive climate, SWEC = Success-without-effort climate; ** *p* < 0.01 and * *p* < 0.05 for statistical significance.

## Data Availability

The data presented in this study are available on request from the corresponding author due to the confidentiality and anonymity of participants’ responses.

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
