# Peer review of "Unveiling Players’ Perceptions of Mother- and Father-Initiated Motivational Climates and Fear of Failure in Youth Male Team Sports"

_sports, 2024, doi:10.3390/sports12090244_

Round 1

Reviewer 1 Report

Comments and Suggestions for Authors

Dear Editor,

Thank you for the opportunity to review this manuscript for Sports. The paper titled "Unveiling players' perceptions of mother and father-initiated motivational climate and fear of failure on youth male team sports" examines the relationship between perceived parental motivational climate and fear of failure in adolescent male team sport athletes. The main findings indicate that father-initiated motivational climate had a stronger association with fear of failure compared to mother-initiated climate, and that mastery-oriented climates were negatively associated with fear of failure while ego-oriented climates were positively associated. Overall, this manuscript presents an interesting and valuable contribution to the literature on parental influences in youth sport psychology. With some revisions to strengthen the methodology and discussion of implications, it has the potential to be a strong publication.

General comments:

1.     The introduction provides a good overview of the relevant theoretical background, but could be strengthened by more clearly articulating the specific gaps in the literature that this study aims to address. The novelty and significance of examining mother vs. father climate separately, and looking at different team sports, should be emphasized more.

2.     The methodology section would benefit from more detail on participant recruitment and data collection procedures. Information on response rates, potential selection bias, and steps taken to ensure data quality should be included.

3.     The statistical analyses are appropriate, but more justification should be provided for the specific analyses chosen. Additionally, effect sizes should be reported consistently throughout the results.

4.     The discussion does a good job of interpreting the main findings, but could be expanded to more thoroughly address the practical implications of the results for parents, coaches, and sport organizations. Concrete recommendations should be provided.

5.     The limitations section is brief and could be expanded to more critically evaluate potential weaknesses in the study design and generalizability of findings.

Specific comments:

Introduction:

·       The statement "Studies have consistently demonstrated the positive impact of parent-initiated mastery-oriented motivational climate on athlete's overall development" needs citations to support it.

·       The paragraph on the rationale for studying team sports is important but could be expanded to more clearly justify why examining different team sports adds value.

Methods:

·       More details are needed on the informed consent procedures, particularly for underage participants. Was parental consent obtained?

·       The assumptions for multiple linear regression should be stated more explicitly (e.g., normality, homoscedasticity, multicollinearity).

Results:

·       Effect sizes (e.g., r2) should be reported in addition to correlation coefficients in Table 1.

·       The statement about handball and water polo players needs more explanation or context.

·       Table 4 contains a large amount of data that is difficult to interpret. Consider presenting key findings in a more accessible format, perhaps with visualization.

Discussion:

·       The limitations section should also address potential issues with self-report measures and social desirability bias.

·       The conclusion could be strengthened by offering more specific, actionable recommendations for how the findings can be applied by parents, coaches, and sport organizations.

In summary, this manuscript presents valuable data on an important topic in youth sport psychology. With revisions to strengthen the methodology reporting, expand on practical implications, and enhance the overall clarity and organization, it has the potential to make a strong contribution to the field.

Author Response

Comment 1: The introduction provides a good overview of the relevant theoretical background but could be strengthened by more clearly articulating the specific gaps in the literature that this study aims to address. The novelty and significance of examining mother vs. father climate separately, and looking at different team sports, should be emphasized more.

Response 1: Thank you for pointing this out. We proceeded with the revision accordingly [page 3, lines 114-122].

Comment 2: The methodology section would benefit from more detail on participant recruitment and data collection procedures. Information on response rates, potential selection bias, and steps taken to ensure data quality should be included.

Response 2: We revised the methodology section according to these suggestions [throughout the methodology section].

Comment 3: The statistical analyses are appropriate, but more justification should be provided for the specific analyses chosen. Additionally, effect sizes should be reported consistently throughout the results.

Response 3: Thank you for pointing this out. The justifications concerning the specific analyses chosen were already clearly explained in the text. We reported the effect sizes the respective results [throughout the results section].

Comment 4: The discussion does a good job of interpreting the main findings, but could be expanded to more thoroughly address the practical implications of the results for parents, coaches, and sport organizations. Concrete recommendations should be provided.

Response 4: we provided concrete recommendations for parents, coaches and sport organizations [page 11, lines 419-426 and pag2 12, lines 447-453].

Comment 5: The limitations section is brief and could be expanded to more critically evaluate potential weaknesses in the study design and generalizability of findings.

Response 5: We revised the text accordingly [page 11, lines 386-426].

Comment 6: The statement "Studies have consistently demonstrated the positive impact of parent-initiated mastery-oriented motivational climate on athlete's overall development" needs citations to support it.

Response 6: we included the respective citations [page 2, line 45]

Comment 7: The paragraph on the rationale for studying team sports is important but could be expanded to more clearly justify why examining different team sports adds value.

Response 7: we provided this justification [page 3, lines 104-109].

Comment 8: More details are needed on the informed consent procedures, particularly for underage participants. Was parental consent obtained?

Response 8: we included this information [page 4, line 153].

Comment 9: The assumptions for multiple linear regression should be stated more explicitly (e.g., normality, homoscedasticity, multicollinearity).

Response 9: we revised the text accordingly [pages 5, lines 207-211].

Comment 10: Effect sizes (e.g., r2) should be reported in addition to correlation coefficients in Table 1.

Response 10: Thank you for pointing this out. We understand your point of view, however the Pearson product-moment correlation coefficient is measured on a standard scale and as such we can interpret the correlation coefficient as representing an effect size. It tells us the strength of the relationship between the two variables. We believe that in this case adding the R2 it will be redundant.

Comment 11: The statement about handball and water polo players needs more explanation or context.

Response 11: The explanation of the results about handball and water polo players are fully explored and explained in the discussion section [Page 10, lines 359-371].

Comment 12: Table 4 contains a large amount of data that is difficult to interpret. Consider presenting key findings in a more accessible format, perhaps with visualization.

Response 12: We understand your point of view and appreciate your comment. Although the table is dense, it contains all the essential information regarding the multiple linear regression analysis for Parent-Initiated Motivational Climate (mother and father) and Fear of Failure, categorized by type of sport. This information is crucial for understanding the results.

Comment 13: The limitations section should also address potential issues with self-report measures and social desirability bias.

Response 13: We addressed this issue on the limitation sections [page 11, lines 393-396].

Comment 14: The conclusion could be strengthened by offering more specific, actionable recommendations for how the findings can be applied by parents, coaches, and sport organizations.

Response 14: We included this information both at the end of discussion and conclusion [page 11, lines 419-426 and page 12, lines 447-453].

Reviewer 2 Report

Comments and Suggestions for Authors

General Comments:

The manuscript addresses a significant topic in the field of sports psychology, with a particular focus on the impact of parental motivational climates on the fear of failure among youth athletes. However, certain aspects require further refinement prior.

Introduction

The introduction is generally well written; however, the authors must provide a more detailed explanation of how mother- and father-initiated motivational climates can influence fear of failure in male youth athletes engaged in team sports. The current format does not explicitly describe the relationship from motivational climate to fear of failure.

Furthermore, the authors must provide a rationale for their decision to focus on youth male team sports. Please clarify why female athletes and individual sports were excluded from the study.

Materials and Methods

The content of lines 126-127 is not sufficiently clear. Please provide more detailed information.

The authors must justify the discrepancies in the sample sizes utilized for their data analysis.

Prior to conducting the regression analysis, it is essential to ascertain whether there are any issues with multicollinearity.

Discussion

Further discussion is required to elucidate why certain sports demonstrate stronger correlations between parental climate and fear of failure. A more speculative approach to these differences would be beneficial in the discussion section, with potential links to existing literature or suggestions for further research.

The practical implications could be developed further and made more specific to different stakeholders (e.g., coaches, parents, policymakers).

Furthermore, the manuscript should address the limitations of generalizing these findings across diverse cultural and socioeconomic contexts. The manuscript posits that father-initiated climates exert a more pronounced influence on fear of failure but does not delve deeply into the underlying reasons for this phenomenon.

Author Response

Comment 1: The introduction is generally well written; however, the authors must provide a more detailed explanation of how mother- and father-initiated motivational climates can influence fear of failure in male youth athletes engaged in team sports. The current format does not explicitly describe the relationship from motivational climate to fear of failure. 

Response 1: We revised the text accordingly [page 2, lines 76-82 and page 3, lines 104-109]

Comment 2: Furthermore, the authors must provide a rationale for their decision to focus on youth male team sports. Please clarify why female athletes and individual sports were excluded from the study.

Response 2: We included this information [page 3, lines 101-103 and 112-114].

Comment 3: The content of lines 126-127 is not sufficiently clear. Please provide more detailed information.

Response 3: Thank you for pointing this out. However, we believe the manuscript is already extensive and this information is supplementary. The reader can easily find more information in the citation provided. This strategy is largely used in scientific reports, as we can see, for example, in Guimarães et al. 2021. (Guimarães, E., Baxter-Jones, A., Williams, A. M., Tavares, F., Janeira, M. A., & Maia, J. (2021). The role of growth, maturation and sporting environment on the development of performance and technical and tactical skills in youth basketball players: the INEX study. Journal of Sports Sciences, 39(9), 979-991. https://doi.org/10.1080/02640414.2020.1853334 )

Comment 4: The authors must justify the discrepancies in the sample sizes utilized for their data analysis.

Response 4: We included this information [page 3, lines 144-148].

Comment 5: Prior to conducting the regression analysis, it is essential to ascertain whether there are any issues with multicollinearity.

Response 5: We ensured this issue and have explained in the text [page 5, lines 207-211].

Comment 6: Further discussion is required to elucidate why certain sports demonstrate stronger correlations between parental climate and fear of failure. A more speculative approach to these differences would be beneficial in the discussion section, with potential links to existing literature or suggestions for further research.

Response 6: We do not support including speculation in the discussion section. Nevertheless, we have made an effort to enhance the discussion section, as well as the limitations and conclusions, in response to the reviewers' comments.

Comment 7: The practical implications could be developed further and made more specific to different stakeholders (e.g., coaches, parents, policymakers). 

Response 7: We revised the text accordingly [page 11, lines 419-426 and page 12, lines 447-453].

Comment 8: Furthermore, the manuscript should address the limitations of generalizing these findings across diverse cultural and socioeconomic contexts. The manuscript posits that father-initiated climates exert a more pronounced influence on fear of failure but does not delve deeply into the underlying reasons for this phenomenon.

Response 8: We included this information on the discussion section [page 10, lines 337-348]. We also revised the limitation section accordingly [page 11, lines 408-411].

Reviewer 3 Report

Comments and Suggestions for Authors

This is an interesting study that aimed to examine the relationship between perceived mother and father motivational climate and players’ fear of failure.

Initially, I found it ideal that the authors included male players from team sports as the inclusion of individual sports might have resulted in different outcomes.

The abstract is clearly presenting the purpose, methodology as well as the main outcomes of the study. The findings of this study can be used by coaches and sports organizations as well as the athletes' parents.

The introduction is informative and includes all the relevant literature. I find the introduction long but considering that the information presented is relevant to the motivational climate of the players and the fear of failure I wouldn’t change it.

The purpose of the study is clearly presented.

Materials and Methods

I would present the mean and SD with a dot instead of a comma (line 130). Average age 14.14 ± 0.82 years.

The measurement instrument and statistical analysis were presented in detail.

Results

The mean and SD presented below the parameters of interest in Table 1 are a little confusing. Mean and SD could have been presented in a different Table with the Pearson product correlations presented in a different one. This is just an observation, however, as it is a minor issue. If someone reads the Table carefully it makes sense.

The same applies to Table 4 which is difficult to follow. I do understand the difficulty of presenting all the parameters, however.

Discussion

I enjoyed reading the discussion part and the conclusions. I would suggest that parents need to read the results of this study before anyone else.

Comments on the Quality of English Language

The English language is fine. Some minor issues can be easily corrected by a Native English Speaker.

Author Response

Comment 1: I would present the mean and SD with a dot instead of a comma (line 130). Average age 14.14 ± 0.82 years.

Response 1: We revised the text accordingly [page 3, line 136]

Comment 2: The mean and SD presented below the parameters of interest in Table 1 are a little confusing. Mean and SD could have been presented in a different Table with the Pearson product correlations presented in a different one. This is just an observation, however, as it is a minor issue. If someone reads the Table carefully it makes sense.

Response 2: Many thanks for pointing this out. We really believe the table 1 should include all the information. This is the guidance provided by APA and followed by several research article on this topic.

Comment 3: The same applies to Table 4 which is difficult to follow. I do understand the difficulty of presenting all the parameters, however.

Response 3: Many thanks again for the point raised. We understand your point of view and appreciate your comment. Although the table is dense, it contains all the essential information regarding the multiple linear regression analysis for Parent-Initiated Motivational Climate (mother and father) and Fear of Failure, categorized by type of sport. This information is crucial for understanding the results.

Round 2

Reviewer 2 Report

Comments and Suggestions for Authors

The authors have addressed comments, and the manuscript has been sufficiently improved to be considered for publication. Thank the authors for their works on this manuscript.